# Antibiotic Resistance and Genotypes of Nosocomial Strains of *Acinetobacter baumannii* in Kazakhstan

**DOI:** 10.3390/antibiotics10040382

**Published:** 2021-04-03

**Authors:** Alyona Lavrinenko, Eugene Sheck, Svetlana Kolesnichenko, Ilya Azizov, Anar Turmukhambetova

**Affiliations:** 1Share Resource Laboratory, Karaganda Medical University, Karaganda 100008, Kazakhstan; lavrinenko@qmu.kz (A.L.); turmuhambetova@qmu.kz (A.T.); 2Institute of Antimicrobial Chemotherapy, Smolensk State Medical University, 214014 Smolensk, Russia; evgeniy.sheck@antibiotic.ru (E.S.); ilya.azizov@antibiotic.ru (I.A.)

**Keywords:** *Acinetobacter baumannii*, multidrug-resistant bacteria, carbapenemases, nosocomial infection

## Abstract

The aim of this study was to determine the prevalence of *A. baumannii* antibiotic-resistant strains in Kazakhstan and to characterize genotypes related to epidemic “high-risk” clones. Two hundred and twenty four *A. baumannii* isolates from four cities of Kazakhstan in 2011–2019 were studied. Antibiotic susceptibility testing was performed by using broth microdilutions method according to EUCAST (v 11.0) recommendations. The presence of *bla*_OXA-23-like_, *bla*_OXA-24/40-like,_
*bla*_OXA-58-like,_
*bla*_VIM,_
*bla*_IMP,_ and *bla*_NDM_ genes was determined by PCR. Genotyping was performed using high-throughput real-time PCR detection of 21 SNPs at 10 chromosomal loci used in existing MLST schemes. Resistance rates to imipenem, meropenem, amikacin, gentamicin, and ciprofloxacin were 81.3%, 78.6%, 79.9%, 65.2%, and 89.3%, respectively. No colistin resistant isolates were detected. The values of the MIC 50% and the MIC 90% of tigecycline were 0.125 mg/L, only four isolates (1.8%) had the ECOFF value >0.5 mg/L. The presence of acquired carbapenemase genes was found in 82.2% strains, including *bla*_OXA-23-like_ (78.6%) or *bla*_OXA-58-like_ (3.6%) genes. The spreading of carbapenem resistant *A. baumannii* strains in Kazakhstan was associated with epidemic “high-risk” clonal groups, predominantly, CG208(92)^OXF^/CG2^PAS^ (80.8%) and less often CG231(109)^OXF^/CG1^PAS^ (1.8%).

## 1. Introduction

*Acinetobacter baumannii* is an opportunistic pathogen that often causes diseases in immunocompromised patients [1]. *A. baumannii* is common in hospitals and causes a variety of nosocomial infections and iatrogenic diseases that include bloodstream infections, urinary tract infections, meningitis, wound infections, and more [1,2]. The World Health Organization categorizes included MDR *A. baumannii* as the “highest priority pathogen” for which antibiotic development is urgently needed [3]. However, carbapenems are considered as effective antibiotics against many drug-resistant microorganisms [4]. At once, the number of carbapenem-resistant *A. baumannii* isolates has been increasing recently [5].

The prevalence of carbapenem resistance of *A. baumannii* in etiological structure of nosocomial infections during last decade was heterogeneous. According to the EARS-Net data from 2013 to 2017, the part of carbapenem resistant *A. baumannii* in West Europe was 35.6%, at this time the prevalence of carbapenem resistant *A. baumannii* in Greece and Turkey were 79% and 95% respectively. However, according to longitude study in Austria the level of carbapenem resistant *A. baumannii* was more than 97% [6]. In alignment, the CHINET surveillance system in China from 2004 to 2015 the level of carbapenem resistant *A. baumannii* increased from 31 to 66.7% [7]. Russian multicenter surveillance study shown the comparable data when carbapenem resistant *A. baumannii* were detected in 77% [8]. According to multicenter study nosocomial infections caused by *A. baumannii* in Pakistan the rate of carbapenem resistance in 2017 was 89% [9].

Most isolates are resistant to carbapenems, and also become resistant to other classes of antimicrobial drugs such as aminoglycosides, fluoroquinolones, and to polymyxin in sporadic cases [10,11]. The rapid spread of multidrug-resistant nosocomial strains of *Acinetobacter* is a global concern. The present situation determines the need for regular monitoring of the sensitivity of nosocomial strains of *Acinetobacter* spp. and, if necessary, correction of the therapy strategy for infections caused by them [12,13]. The global spread of multidrug resistant strains, including resistant to carbapenems, among nosocomial *A. baumannii* strains was associated with expansion of two international “high-risk” clonal lineages, called ICL1 and ICL2 [14,15,16,17].

Microbial genotyping methods is an important tool for molecular epidemiological studies, particularly, for understanding of the population dynamics and transmission of pathogens. Even the ubiquity of whole-genome data, multilocus sequence typing (MLST) is still a “gold standard” for molecular typing of *A. baumannii* due to standardized landscape about population structure [18,19]. Two available MLST approaches for *A. baumannii* (Oxford [20] and Pasteur [14] schemes) characterized by different resolution ability but provide concordant discrimination at the level of predominant clonal groups [21,22]. In epidemiological studies, the epidemic “high-risk” clones are typically defined using by MLST nomenclature of sequence types (STs) and clonal complexes (CCs) or clonal groups (CGs) [23] and are characterized as CC231(109)^OXF^/CC1^PAS^ (for ICL1) and CC208(92)^OXF^/CC2^PAS^ (for ICL2) [16]. However, such approaches is still expensive and laborious especially for analysis of large sample collections, whereas high-throughput approaches allow to study in detail the entire diversity of *A. baumannii* strains. A lot of existing high-throughput approaches have been proposed [24,25,26,27,28,29,30] but only single nucleotide typing (SNP)-typing scheme provide direct compare SNP-typing and MLST data due to identify a set of SNPs of MLST loci [30].

In the present study, we aimed to investigate the rates of antibiotic resistance and production of acquired carbapenemase genes and to determine the genotypes and prevalence of “international high-risk clones” among *A. baumannii* isolates in Kazakhstan.

## 2. Results and Discussion

Determining the antimicrobial susceptibility of *A. baumannii* strains results are presented in Table 1. Resistance to carbapenems (imipenem and meropenem) were shown by 81.3% and 78.6% of *A. baumannii* isolates, respectively. Comparable the resistance phenotype was detected in South-Est Asia: Korea (87.0%), Singapore (95.2%), Hong Kong (50.0%) and Thailand (59.2%) [31]. Fluoroquinolones have high resistance indicators: 89.3% of *A. baumannii* isolates were resistant to ciprofloxacin. The frequency of resistance to aminoglycosides (amikacin and gentamicin) was 79.9% and 65.2%, respectively. Among non-beta-lactam antibiotics, colistin was highly active in vitro, none resistant isolates were detected. The values of the MIC 50% and the MIC 90% of tigecycline were 0.125 mg/l, only four isolates (1.8%) had the ECOFF value >0.5 mg/L. Combined resistance to imipenem, amikacin, and ciprofloxacin was observed in 73.2% isolates. Extremely profile of resistance make polymyxins (polymyxin B and colistin) as a last-resort treatment option [32].

The presence of genes for acquired molecular class D carbapenemases belonging to the OXA-23 (78.6%) and OXA-58 (3.6%) groups were revealed in 82.2% of *A. baumannii* isolates. No MBL genes were found in *A. baumannii* isolates. Actually, number of isolates harboring carbapenemase genes was higher than carbapenem resistant isolates: eight isolates were susceptible to imipenem and meropenem despite the presence of carbapenemase genes. The OXA-23 carbapenemases are major mechanism of resistance *A. baumannii* to carbapenems in the Central Asia and early the spreading of OXA-23 produced hospitals acquired strains of *A. baumannii* was described in Russia [8] and South-East Asia countries [17]. The results of assessing the sensitivity of *A. baumannii* isolates carrying the genes of acquired OXA-carbapenemases are presented in Table 2. The majority of carbapenemase producers were characterized by associated resistance to ciprofloxacin (97.1%), amikacin (89.7%) and gentamicin (69.5%). No resistant to colistin strains were observed.

All isolates were distributed into 20 genotypes (SNP-types) that grouped into 14 clonal groups (genetic clusters combining strains of related SNP-types) (Figure 1). Genotypes that differed in one or two SNPs were considered related. Despite this diversity, most of isolates (*n* = 181, 80.8%) from four cities were assigned to the same clonal group that combined five related genotypes (SNP-types). Through *in silico* analysis, nucleotide profiles of these genotypes corresponded to a set of STs that combined into a CG208^OXF^ (formerly known as CC92^OXF^) and CG2^PAS^. Strains of clonal group CG208(92)^OXF^/CG2^PAS^ were characterized by a high frequency of *bla*_OXA-23-like_ carbapenemase genes (170 of 181 isolates, 93.9%). CG208(92)^OXF^/CG^2PAS^ is related to international clonal lineage ICL2 that was associated with global dissemination of multidrug resistant *A. baumannii* strains, including neighboring countries [8,16,34].

Among other clonal groups, isolates harboring *bla*_OXA-23-like_ carbapenemase genes have been also found in SNP-type 7 (3 of 4 isolates, 75.0%) and SNP-type 47 (3 of 3 isolates, 100.0%). Through *in silico* analysis, nucleotide profiles of SNP-type 7 corresponded to a set of STs that combined into a CG231^OXF^ (formerly known as CC109^OXF^) and CG1^PAS^. CG231(109)^OXF^/CG1^PAS^ was corresponded to another international clonal lineage ICL1 [16,34], however, an incidence of CG231(109)^OXF^/CG1^PAS^ was sporadic in Kazakhstan—four isolates were collected from Nur-Sultan in 2014 and 2018. Three isolates of SNP-type 47 harboring *bla*_OXA-23-like_ genes were also collected from Nur-Sultan in 2015. Interestingly enough, the epidemic “high-risk clones” CG231(109)^OXF^/CG1^PAS^ and CG208(92)^OXF^/CG^2PAS^ harboring *bla*_OXA-23-like_ carbapenemases genes were concomitant with spreading of carbapenem resistant *A. baumannii* strains in Pakistan [9].

Isolates harboring *bla*_OXA-58-like_ carbapenemase genes (*n* = 8) have been found in same genotype (SNP-type 83). All isolates of this genotype were carriers of *bla*_OXA-58-like_ genes and were isolated in 2016 and 2017 from Nur-Sultan. One isolate of SNP-type 83 was investigated by MLST to clarify phylogenetic traits and was assigned to ST184^OXF^ and ST218^PAS^. Strains of this clonal group CG184^OXF^/CG218^PAS^ were previously found in South Korea in 2008 and in China in 2009-2010, according to the pubMLST database. Strains of CG184^OXF^/CG218^PAS^ have not been previously detected in neighboring countries.

Despite the presence of *bla*_OXA-58-like_ genes, all but one isolates of CG184^OXF^/CG218^PAS^ were susceptible for imipenem and meropenem. For susceptible isolates, distribution of MIC ranged from 0.5 to 1 mg/L for imipenem and 0.25 to 1 mg/L for meropenem. Expression of *bla*_OXA-58-like_ gene depends on the presence of insertion sequence, most often ISAba3, in association with carbapenemase gene [36,37], so, presence of *bla*_OXA-58-like_ gene does not always lead to resistance to carbapenems. Furthermore, another isolate harbored *bla*_OXA-23-like_ gene was found in CG208(92)^OXF^/CG2^PAS^ and had a low MIC values (0.5 mg/L) for imipenem and meropenem.

*A. baumannii* strains harboring *bla*_OXA-58-like_ genes were appeared quite recently in Kazakhstan although that strains were a serious problem since the dawn of the worldwide spreading of carbapenem resistant strains of *A. baumannii* [16]. For the first mention, OXA-58 carbapenemase was found in *A. baumannii* isolate in France in 2003 [38]. Afterwards, *A. baumannii* strains harboring *bla*_OXA-58-like_ genes became widespread in European (Greece, France, Belgium, Italy and Turkey [39,40,41,42]), American (Argentina and USA [43,44,45]), and Asian regions (China, Taiwan and Singapore [46,47,48]). Patient transportation appears to be one of decisive factors in the spread of OXA-58-positive strains in global population of *A. baumannii* [40,44]. In neighboring countries, OXA-58-positive strains were found in Russia and that strains were noticed in cities bordering Kazakhstan, including Novosibirsk and Yekaterinburg (https://amrmap.net/, last accessed on 12 March 2021 [49]) [8]. According some epidemiological studies, several countries (China, Austria and Iran) remains the endemic regions of *A. baumannii* strains harboring *bla*_OXA-58-like_ genes [6,50,51]. Thus, spread of OXA-58-producing strains in the *A. baumannii* population fell on 2005–2010, whereas now there has been a shift towards carbapenemase of the OXA-23-like and OXA-24/40-like [48,52,53].

Comparison of resistance among isolates of different clonal groups is presented in Table 3. CG208(92)^OXF^/CG2^PAS^ isolates were significantly more resistant to multiple drugs compared to minor genotypes: to imipenem (93.9% vs 22.6%, *p* < 0.0001), meropenem (93.4% vs 12.9%, *p* < 0.0001), amikacin (91.7% vs 25.8%, *p* < 0.0001), gentamicin (68.5% vs 32.3%, *p* = 0.0002), and ciprofloxacin (100.0% vs 38.7%, *p* < 0.0001). It was impossible to compare resistance for CG231(109)^OXF^/CG1^PAS^ or CG184^OXF^/CG218^PAS^ due to small number of strains. The values of MIC 90% for tigecycline were 0.125 mg/L for each clonal groups while isolates with ECOFF value >0.5 mg/L were found in CG208(92)^OXF^/CG2^PAS^ (*n* = 3) and SNP-type 37 (*n* = 1). Combined resistance to imipenem, amikacin, and ciprofloxacin among CG208(92)^OXF^/CG2^PAS^ isolates was also significantly (*p* > 0.0001) higher compared to minor genotypes: 87.3% (95% CI: 81.7.91.4%) vs 12.9% (95% CI: 5.1–28.9%). Two of four isolates of CG231(109)^OXF^/CG1^PAS^ and none of CG184^OXF^/CG218^PAS^ also had combined resistance to imipenem, amikacin and ciprofloxacin.

Thus, the results of this study indicate the advisability in the routine microbiological diagnostic the detection of carbapenemases genes and genetic profiles for phylogenetic and epidemiological surveillance.

## 3. Materials and Methods

### 3.1. Sources of Bacterial Isolates

*Acinetobacter baumannii* (*n* = 224) were collected from inpatients of four cities of Kazakhstan (Karaganda, Nur-Sultan, Almaty, and Jezkazgan) in 2011–2019. Isolation and primary identification of bacterial isolates were carried out in local clinical microbiological laboratories by using standard microbiological methods. The final identification of all bacterial isolates was made in the share resource laboratory of the Karaganda Medical University (Kazakhstan). Molecular genetic studies were performed in the Research Institute of Antimicrobial Chemotherapy (Smolensk, Russia).

Material obtained from patients after 48 h of admission to the dispensary for confirmation of nosocomial infection [54,55]. Collected isolates were recovered from respiratory tract of intensive care unit patients (*n* = 126, 56.7%), surgical intra-abdominal infection (*n* = 35, 15.6%), blood of intensive care patients (*n* = 18, 8.0%), urinary tract infections (*n* = 17, 7.6%), surgical soft tissue infections (*n* = 17, 7.6%), bones and joints infections (*n* = 3, 1.3%), central nervous system lesions (*n* = 3, 1.3%), and other sources (*n* = 4, 1.8%).

### 3.2. Species Identification and Storage of Isolates

The isolates were identified by matrix-assisted laser desorption/ionization—time-of-flight mass spectrometry (MALDI-TOF MS) using the Microflex LT system and the MALDI Biotyper Compass v.4.1.80 software (Bruker Daltonics, Hamburg, Germany). The values known for *Acinetobacter* representatives were used as a criterion for reliable species identification score ≥ 2.0 according to manual. The species identification of *A. baumanii* isolates was confirmed by detection of species-specific *bla*_OXA-51-like_ genes using real-time PCR with commercial kits “AmpliSensR MDR Ab-OXA-FL” (Central Research Institute of Epidemiology, Moscow, Russia) and the DTPrime 5X1 system (DNA-Technology, Moscow, Russia). Prior to analysis, the isolates were stored at −70 °C in trypticase-soy broth (BD, Sparks, MD, USA) supplemented with 30% glycerol. The obtained strains from local laboratories were revived on Columbia blood agar (BD, USA) aerobically at 36 ± 1 °C.

### 3.3. Determination of Sensitivity to Antibiotics

Determination of sensitivity to antimicrobial drugs (amikacin, gentamicin, imipenem, meropenem, netilmicin, ciprofloxacin, tigecycline, colistin) was carried out by the microdilution method in Mueller-Hinton broth. Interpretation of susceptibility testing results was performed according to recommendation of EUCAST v 11.0 [33]. To control the quality of the sensitivity determination, we used *Escherichia coli* ATCC^®^25922, *Escherichia coli* ATCC^®^35218 strains, *Pseudomonas aeruginosa* ATCC^®^27853.

### 3.4. Identification of Carbapenemase Genes

The presence of acquired class D carbapenemases genes common for *Acinetobacter* spp. (groups OXA-23, OXA-24/40, and OXA-58), as well as class B carbapenemases (metallo-beta-lactamases (MBL) of VIM, IMP, and NDM groups) were determined by real-time PCR using commercial kits “AmpliSensR MDR Acinetobacter-OXA-FL” and “AmpliSensR MDR MBL-FL” (Central Research Institute of Epidemiology, Moscow, Russia). For amplification, a DTPrime 5X1 real-time PCR system (DNA Technology, Moscow, Russia) was used. Strains *A. baumannii*, *A. pittii*, and *P. aeruginosa*, carring the known carbapenemases genes of the listed groupswere used as positive controls. DNA extraction was performed by express method using InstaGeneTM matrix (Bio-Rad, Hercules, CA, USA). Samples of extracted DNA were stored at −20 °C before testing. The results of assessing the sensitivity to antibiotics and determining the genes of various types of carbapenemases have been deposited to the AMRmap website database [49].

### 3.5. Molecular Genotyping of A. baumannii

Genetic diversity of *A. baumannii* was explored by single nucleotide polymorphism (SNP)-typing method based on the analysis of 21 informative SNPs at 10 chromosomal loci (*gltA*, *recA*, *cpn60*, *gyrB*, *gdhB*, *rpoD*, *fusA*, *pyrG*, *rplB*, and *rpoB*) used in the University of Oxford and the Institute Pasteur multilocus sequencing-typing (MLST) schemes [30]. Detection of each SNP was performed by allele-specific real-time PCR according to the high-throughput approach proposed Myakishev et al. [56]. Obtained sequences of 21 nucleotide bases for each *A. baumannii* isolate were used to assign a certain genotype (SNP-type) and to assess genetic relatedness between genotypes. The QIAgility system (QIAGEN, Hilden, Germany) and DTPrime 5X1 (DNA-Technology, Moscow, Russia) were used to prepare and conduct PCR in 384-well format that ensured high-throughput genotyping of isolates. *A. baumannii* strains of known STs submitted to pubMLST database (https://pubmlst.org/organisms/acinetobacter-baumannii, last accessed on 12 March 2021): id4785, id4790, id4793, id4798, id4816, id4825, id4841, id4932, id4985, id5036 were used as controls.

The selected set of 21 SNPs from MLST loci provided a comparison between obtained SNP-types with known STs and CCs according to the MLST nomenclature, including the so-called “high-risk international clones”. That correspondence between SNP-typing and MLST data was provided by SQL database and software platform (called SNPTAb, http://snpt.antibiotic.ru:9002/, last accessed on 12 March 2021) [57]. Furthermore, the SNPTAb database was used to store of SNP-typing data with individual isolates data (e.g., source, geographical origin, data of isolation, resistance to carbapenems, and production of carbapenemases). Cluster analysis of obtained SNP profiles and was carried out using the PHYLOViZ 2.0 software [35].

MLST was performed for one isolate using both the University of Oxford and the Institute Pasteur schemes as described previously [14,20]. Obtained MLST sequences were uploaded to PubMLST database (https://pubmlst.org/organisms/acinetobacter-baumannii, last accessed on 12 March 2021) to identify alleles and STs.

## 4. Conclusions

Nosocomial infections by carbapenem resistant *A. baumannii* strains emerge sharply in Kazakhstan since 2011. The results of this study indicate a high prevalence of resistance to most antimicrobials, including all carbapenems, aminoglycosides, and fluoroquinolones used to treat infections caused by this pathogen. Colistin was highly active on all *A. baumannii* isolates. The main mechanism of resistance to carbapenems of *A. baumannii* was the production of acquired carbapenemases belonging to the OXA-23 group. The spreading of carbapenem resistant *A. baumannii* strains in Kazakhstan was associated with epidemic “high-risk” clonal groups, predominantly, CG208(92)^OXF^/CG2^PAS^ (80.8% isolates) and less often CG231(109)^OXF^/CG1^PAS^ (1.8% isolates). Isolates of these clonal groups were significantly more resistant to aminoglycosides and fluoroquinolones compared to other genetic lines. Furthermore, several isolates were carrying *bla*_OXA-58-like_ carbapenemase genes and combined into one clonal group CG184^OXF^/CG218^PAS^. Given the high probability of *A. baumannii* strains resistance to the main antibacterial drugs for the treatment of nosocomial infections, the choice of antibiotics for empiric therapy is extremely difficult and requires regular local monitoring of sensitivity in each hospital.

## Figures and Tables

**Figure 1 antibiotics-10-00382-f001:**
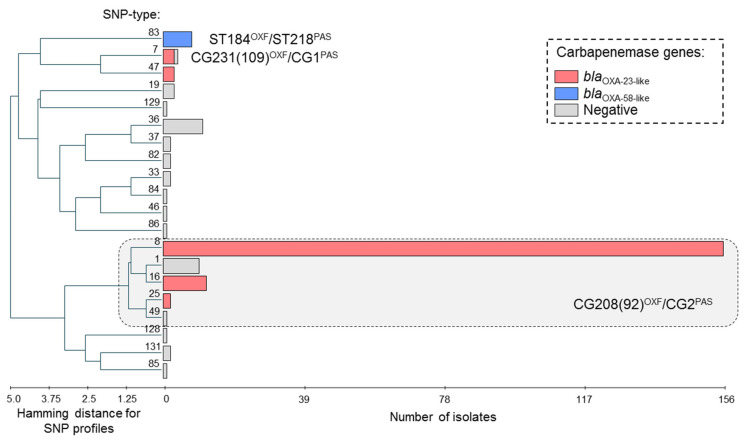
Genetic diversity and production of carbapenemases in *A. baumannii* strains in Kazakhstan. UPGMA algorithm was used for hierarchical cluster analysis by PHYLOViZ software [35]. Horizontal rectangles correspond to different genotypes (SNP-types). Length of the rectangles is proportional to the number of isolates. MLST nomenclature was used to characterize of prevalent clonal groups. Types of acquired carbapenemase genes are highlighted.

**Table 1 antibiotics-10-00382-t001:** Antibiotic sensitivity of *A. baumannii* isolates (*n* = 224).

Name of Antibiotic	% of Isolates and MIC Value, mg/L													% of Isolates by Category			MIC, mg/L	
	≤0.06	0.125	0.25	0.5	1	2	4	8	16	32	64	128	≥256	S	I	R	50%	90%
Amikacin				3.6	1.8	4.5	6.7	3.6	8.0	20.1	20.1	8.5	23.2	20.1		79.9	64	≥256
Gentamicin			0.4	8.9	5.4	8.0	12.1	11.6	11.6	17.9	7.1	3.6	13.4	34.8		65.2	16	≥256
Imipenem				13.4	4.9		0.4	2.2	21.4	29.9	22.8	3.1	1.8	18.3	0.5	81.3	32	64
Meropenem			0.4	15.2	3.6	0.4		1.8	32.6	28.6	13.8	1.8	1.8	19.6	1.8	78.6	16	64
Netilmicin				7.1	11.2	18.8	15.6	8.0	14.7	11.2	5.4	1.8	6.3	ND	ND	ND	4	64
Ciprofloxacin				6.3	4.5	2.2	0.4	0.4	0.4	0.0	3.6	2.2	79.9	0.0	10.7	89.3	≥256	≥256
Tigecycline	32.6	62.1	0.9	2.7	1.8									ND	ND	ND	≤0.06	0.125
Colistin				18.3	79.9	1.8								100.0		0.0	1	1

ND—no data, according to EUCAST criteria [33] for netilmicin and tigecycline interpretation criteria are absent.

**Table 2 antibiotics-10-00382-t002:** Antibiotic sensitivity of *A. baumannii* isolates (*n* = 184) producing acquired OXA-carbapenemases.

Name of Antibiotic	% of Isolates and MIC Value, mg/L													% of Isolates by Category			MIC, mg/L	
	≤0.06	0.125	0.25	0.5	1	2	4	8	16	32	64	128	≥256	S	I	R	50%	90%
Amikacin				1.1	0.5	2.2	3.8	2.7	8.7	23.9	22.8	10.3	23.9	10.3		89.7	64	≥256
Gentamicin			0.5	4.3	3.3	9.2	13.0	13.6	13.6	19.0	7.1	2.7	13.6	30.4		69.6	16	≥256
Imipenem				2.7	2.2			0.5	26.1	36.4	26.6	3.3	2.2	4.9	0.0	95.1	32	64
Meropenem			0.5	3.3	0.5			2.2	39.1	34.2	16.3	1.6	2.2	4.4	2.2	93.5	32	64
Netilmicin				4.9	8.2	20.1	15.2	8.7	16.3	11.4	6.5	1.6	7.1	ND	ND	ND	8	64
Ciprofloxacin				1.6	1.1	1.1		0.5	0.5		3.3	2.2	89.7	0.0	2.7	97.3	≥256	≥256
Tigecycline	39.7	54.3	1.1	3.3	1.6									ND	ND	ND	≤0.06	0.125
Colistin				13.6	84.8	1.6								100.0		0.0	1	1

ND—no data, according to EUCAST criteria [33] for netilmicin and tigecycline interpretation criteria are absent.

**Table 3 antibiotics-10-00382-t003:** Antibiotic resistance among isolates of different clonal groups.

Antibiotic	CG208(92)^OXF^/CG2^PAS^		CG231(109)^OXF^/CG1^PAS^		CG184^OXF^/CG218^PAS^		Minor Genotypes	
	% of R	95% CI	% of R	95% CI	% of R	95% CI	% of R	95% CI
Imipenem	93.9	89.5–96.6	100.0	51.0–100.0	12.5	2.2–47.1	22.6	11.4–39.8
Meropenem	93.4	88.8–96.2	75.0	30.1–95.4	0.0	0.0–32.4	12.9	5.1–28.9
Amikacin	91.7	86.8–94.9	50.0	15.0–85.0	37.5	13.7–69.4	25.8	13.7–43.3
Gentamicin	68.5	61.4–74.8	100.0	51.0–100.0	100.0	67.6–100.0	32.3	18.6–49.9
Ciprofloxacin	100.0	97.9–100.0	100.0	51.0–100.0	37.5	13.7–69.4	38.7	23.7–56.2

## Data Availability

The data presented in this study are available in the article.

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
