# Peer review of "Antibiotic Resistance and Genotypes of Nosocomial Strains of Acinetobacter baumannii in Kazakhstan"

_antibiotics, 2021, doi:10.3390/antibiotics10040382_

Round 1

Reviewer 1 Report

The manuscript of A. Lavrinenko et al. titled "Antibiotic resistance and genotypes of nosocomial strains of Acinetobacter baumannii in Kazakhstan" describes the antibiotic resistance of nosocomial A. baumannii in three cities of Kazakhstan. The authors used the broth microdilution method for the assessment of MIC of 10 antibiotics.

Major issues:

The manuscript was written in a very unclear manner and very poorly written. Legends for tables do not have all the needed information on what color schemes and what size of circles mean. Since the manuscript was written by non-English-speaking authors, I would highly recommend significant professional editing of the English.
The introduction part is too concise and discussion is absent in the Results and Discussion part.
I would know the comparison of antibiotic resistance/genotyping from previous years and with studies from adjacent Middle Asian countries (Turkmenistan, Uzbekistan, Tajikistan, Kyrgystan), China, and Russia.
Since the manuscript is less than 3,000 words and references are less than 30, I would highly recommend resubmitting it as a Short Communications.

Minor issues:

Line 16: RT-PCR stands for Reverse Transcriptase PCR, not Real-Time PCR.
Line 18: Not clear sentence "78.16% strains showed the presence of genes OXA-23 (78.35%) and OXA-58 (4.12%) groups." Rewrite, please. What percents mean?
Line 20: 1.04% is not the most one.
Line 25: Not good keyword "genes of carbapenemases".
Introduction: Extremely short, only one paragraph.
Line 44: No cefepime in Table 1.
Line 45: 256 and 1024 mg were for both antibiotics (ceftazidime, cefepime)?
Line 49: 83.55% in the text and 83.6% in Table 1.
Lines 43-52: All paragraph is just a repeat of what is in Table 1.
Line 50: For gentamicin, 42.5% in the text and 45.5% in Table 1.
Table 1: MIC, mg/ml, but mg/L in the text (lines 45, 48, 52).
Table 1: Total percentage of isolates for tigecycline is 93.2%. What about the other 6.8%?
Line 65: change to the correct name "gentamicin".
Line 69: In Figure 1 not all isolates (n=435), but only 263.
Line 70: Not clear, what you mean by genotypes? Sequence types, SNP types, or something else?
Line 70: I see not 2 clonal groups, but 3 in Figure 1: CC208OXF, CC231OXF, and CC1507OXF. And what means ~CC1507? Did you really perform any MLST (OXF and PAS), because there is nothing about it in Materials and Methods?
Line 72: CC92 from which MLST scheme, if it CC231OXF/CC1PAS?
Line 74: What genotype and which city you mean?
Figure 1: What numbers mean? Genotypes? SNP types or STs?
Line 90: How do you know STs if you are not performed MLST? If you have done MLST, why it is not mentioned in the Materials and Methods?
Line 102: Colistin and tigecycline were active against which isolates of CC109 / 231OXF / CC1PAS and CC92 / 208OXF / CC2PAS clonal lines?
Line 106: Remove MIC.
Line 111: Could you describe in detail how exactly primary identification of isolates was performed?
Lines 114-115: You can omit mention of colleagues' participation since it is evident for the author's affiliations. The research article is not a place for promotion.
Line 117: Which criteria exactly?
Figure 2: Two slices in the piechart does not have designations: bright green and aquamarine.
Line 131: Manufacturer of trypticase-soy broth?
Line 135: Which antimicrobial drugs?
Line 136: Manufacturer of Mueller-Hinton broth?
Line 137: MIC, since you already explained the acronym in line 45.
Line 138: The EUCAST 9.0 is too outdated, the current version is 11.0. Currently, there is no I category.
Line 139: Escherichia.
Line 141: It is better to cite reference for AMRcloud (AMRcloud: a new paradigm in the monitoring of antibiotic resistance. Clinical Microbiology and Antimicrobial Chemotherapy. 2019; 21(2):119-124).
Line 174: Provide strain numbers for reference strains, please.
Lines 197-203: You can put only initials instead of full names in Author Contributions.
References: Please, add doi numbers for references where it is possible.
References 19-20: Although it is allowable to cite proceedings, you can't cite posters instead of proceedings for references. Please, provide references to proceedings, not to posters.

Reviewer 2 Report

The article is well written, short and reliable. In this form, it fulfills the task of a research article. I only have a few minor comments:

Abstract: It is better to use full names in the abstract; I suggest replacing MDR and MLST abbreviations with them

Introduction: Acinetobacter spp. - spp. without Italic

Figure 1. I did not find in the methodology of description of programs and algorithms according to which the scheme was created. Additionally, it has been characterized in only one sentence. I propose to add a description of the methodology and in the caption to the Figure a deeper description of what the individual digits mean and how to interpret the distances and their mutual arrangement.

Figure 2. I propose to delete this figure and leave only a description. Currently, it is not readable and duplicates the data from the results.

Reviewer 3 Report

The manuscript is focusing on drug resistance and genetic survey of Acinetobacter strains isolated from some clinics and hospitals in Kazakhstan. This work may provide some useful information as a recent Acinetobacter’s epidemiology study. However, some additional information with improvement of the method section is necessary.

Major comments:

  1. According to Line13, these strains were isolated during 2017 to 2020. This should be also described in the method section.
  2. Do these collected strains include outbreak-strains in same clinic or hospital? This should be mentioned in text.
  3. The authors identified these strains using a mass spectrometry method. Are all of strains indeed “baumannii” not “Acinetobacter complex”? In many cases, non-baumanii strains may be found.
  4. The authors used commercial kits from FBIS Central Research Institute of Epidemiology. Is this Russian company??? It is hard for readers to obtain detailed information, such as primers sequence and PCR protocol. Therefore, these information should be described in text and additional table for primers sequence.
  5. Why did not the authors look for genes which are responsible for quinolone and aminoglycoside resistance as well as carbapenemase genes?
  6. Are carbapenamase genes encoded in an exogenous plasmid or in a transposon inserted into the chromosome? Is there possibility that these resistance genes is transferred into other strains? The authors should provide some data or statement.

Minor comments:

Line16: RT-PCR should be changed to real-time PCR because many readers guess that RT-PCR means “Reverse transcription PCR”.

Line56-57: 25% of A. baumannii……colistin. I do not agree this sentence because all strains are also susceptible to tigecycline, right?

Reviewer 4 Report

The paper by Lavrinenko and collaborators aims at the determination of the prevalence of antibiotic resistant strains in 435 isolates of Acinetobacter baumannii (various kinds of infection) collected from 3 clinics located in 3 important cities of Khazakhstan, in relation with their genotypes and more particularly, their belonging to the “high risk international clones”.

This study is of interest for further potential meta-analyses that will follow the progression of those high-risk international clones and, as stated by the authors, to warn the clinicians from Khazakhstan about the necessity to monitor the sensitivity of local isolates to antibiotics and adjust the therapy, if needed.

However, to my point of view, this paper needs to be optimized before publication. The paper is not precise enough and as such, very hard to follow.

1-The introduction is very, very limited. A few more words on the interest of A. baumannii, where it is present, what it leads to are necessary. Same thing for the emergence and spread of the resistances.  Please remind the reader of the criteria used to define a nosocomial infection.

2-It is extremely difficult to understand what are those isolates that were used for the study as there is no data to describe them in the results section. The text of a publication should be reported as a story, which should provide all the data required to understand the figures and tables and conversely, a figure/table should be understandable with the sole help of its legend and without the help of the text. I thus believe, a first paragraph that will describe the isolates, where (city, climate, other relevant description of the region), when they were collected, in which conditions, and from which kind of patients (gender/age/particular clinical status. In particular for the patients with respiratory infection, were they under lung ventilation?) should be added in the text of the results. A few of these data are included in the Material and Methods section … that most readers will not read, leaving them very confused and very frustrated, if the paper is published as such.

3- Table 1: the right column reports MIC50% and MIC90% values in mg/mL while the text that describes these values report them in mg/L! Very confusing! The tested antibiotics should be grouped in families so that the text matches the table in a clearer way. The text mentions the effect of cefepime, which is not included in the table. The second column reports % of isolates, according to their MIC value: which kind of MIC (MIC50 or MIC90)? Line 56: “25% of the how many? tested isolates”…

4- Lines 58-61: very hard to follow as there is no explanation on the choice of the selected genes, no reference, no technique mentioned. Again write it as a story so that the reader is informed about the progression of the reasoning. I thus do not understand those percentages.

5- Table 2: same remarks as for Table 1. Make sure also that this table is not split between 2 pages in the final draft. Another column with the OAXA results would make sense.

6- Figure 1: again, the labelling of the clonal groups in the text does not match with that in the figure. A legend explaining how the grouping of isolates was performed and what the numbers do mean is necessary. As it is, this figure is almost not understandable.

-Line 91: “the genetic line was characterized by a high frequency of resistance to antibiotics in different groups”. What are these different groups? I do not understand.

-Line 106: “netilmicin, MIC tigecycline and colistin”. Why this MIC here?

In short, this paper requires serious rewriting to make it accessible to a large panel of scientific readers.

Round 2

Reviewer 1 Report

After resolving all major and most of the minor issues, the manuscript has been greatly improved by the authors and can be accepted for publication.

The only suggestions for minor improvements are the next one:

Line 40: "more than 97%"
Line 59: Italisize "A. baumannii"
Tables 1 and 2: Add categories for netilmicin and tigecycline
Figure 1: Which software was used for the construction of the UPGMA tree?
Line 151: "France"
Line 187: "(Smolensk, Russia)"
Lines 237 and 252: "STs" instead of "sequence types"
Line 241: Remove "sequence types"
Line 248: It is still not clear, how is in silico MLST was done if Sanger sequencing was not carried out for all strains.
Line 248: For which one isolate MLST was performed? How it was chosen?
Line 250: If in silico MLST was already done using the PHYLOVIZ 2.0 above (line 248), for what reason sequences were uploaded to identify alleles and STs?
Line 360: "2015"?
Lines 392 and 401: pages?
Lines 405 and 411: Volumes and pages?

Author Response

Dear reviewer, thank you for your notes and comments. The manuscript was revised in accordance to your comments. 

Reviewer 3 Report

This manuscript is largely improved. There is no suggestion from me.

Author Response

Thank you!